# Variations in Concentrations and Ratio of Soluble Forms of Nutrients in Atmospheric Depositions and Effects for Marine Coastal Areas of Crimea, Black Sea

Alla V. Varenik * and Sergey K. Konovalov

Marine Biogeochemistry Department, Marine Hydrophysical Institute of RAS, Kapitanskaya Str.,
299011 Sevastopol, Russia; director@mhi-ras.ru
* Correspondence: alla.varenik@mhi-ras.ru

**Abstract:** Atmospheric depositions have been recently recognized as an important source of nutrients for off-shore marine systems, in line with the coastal input and physical exchange. The input of nutrients with atmospheric depositions can change their inventory and ratio in the euphotic zone, thus increase the rate of primary production and the type of predominant phytoplankton. The influence of atmospheric depositions, temporal variations of this influence and consequences of this deposition have been neglected. Monitoring of nutrients in atmospheric depositions of Crimea in 2015–2020 has allowed studying of multi-scale variations in their input to coastal areas and scaling the effects of this input. It has been found that the contribution of dry deposition in the total flux of nutrients is more significant for silicates and phosphates. Intra-annual variations in concentrations of nitrogen reveal a maximum in an urbanized area for the cold period of year, due to burning of extra fuel. On the contrary, increasing concentrations of nitrogen have been detected in a rural area in warm period. High values of concentrations of phosphorus and silica are typical for dry summer period and associated with dust transport from natural and anthropogenic sources. The N:P:Si ratio in the atmospheric depositions has been significantly shifted towards nitrogen as compared to the stoichiometric ratio. The results obtained in this work suggest that additional flux of nutrients with atmospheric depositions is minor at the annual scale, but it may change the local inventory and C:N:P ratio in the surface layer of the sea on a daily-time scale. The input of nutrients with atmospheric depositions can lead to additional (up to twofold) production of organic matter and result in additional oxygen consumption, when this surplus organic matter sinks and is oxidized, thus supporting suboxic conditions in near-shore areas.

**Keywords:** atmospheric depositions; nutrients; inorganic nitrogen; inorganic phosphorus; silicon; Black Sea; ratio of nutrients; oxygen consumption

## 1. Introduction

Information about the sources of nutrients for aquatic ecosystems is of key importance for scaling the level of productivity in a particular water system. The presence and quantity of the main inorganic nutrients (inorganic nitrogen and phosphorus, as well as silicon) along with the intensity of light determines the growth and development of various types of phytoplankton in marine ecosystems, production of organic carbon, consumption of oxygen, when this organic matter sinks and is oxidized below the euphotic zone. This is specifically important for coastal areas because of higher production of organic matter in strong phytoplankton blooms and limited inventory of oxygen below the euphotic zone due to shallow waters. Any additional source of nutrients may support and result in oxygen deficit and suboxic conditions on a short time scale.

Traditionally, the coastal runoff, including riverine input, and the inflow from deeper water layers by upwelling and mixing processes, are considered as the main sources of nutrients for the productive layer of the sea. However, the characteristics (distribution and

capacity) of these sources often cannot explain the peculiarities of the balance of nutrients and production characteristics of a particular water area. The transfer of nutrients with air masses and their input with atmospheric precipitations is one of important but less studied processes even at the qualitative level, not to mention its quantitative and spatio-temporal characteristics.

Various substances and elements are loaded with atmospheric depositions [1–5] onto the underlying surface. Depositions from the atmosphere with snow, fog, or rain are called wet depositions, while depositions as dry particles or gases are called dry deposition. The most influential elements loaded with atmospheric depositions to aquatic systems are nutrients, such as nitrogen in the form of nitrate and ammonium, phosphorus in the form of phosphate and silicon in the form of silicates.

The presence of nutrients fuels production of organic matter, while their ratio influences the composition of produced organic matter. They either critically support the export production in offshore areas, where other sources of nutrients are limited, or provide additional input to coastal areas, where other natural and/or anthropogenic sources have already resulted in intensive eutrophication and lead, in particular, to deoxygenation of the marine environment, increased prevalence of hypoxia and anoxia, and fish kills [6]. Excessive deposition of nutrient may result in algal blooms. Atmospheric depositions may also alter the ratios of nutrient, which in turn may affect compositions of blooming algal species. The ultimate cost of continued high deposition rates may be long-term ecosystem degradation.

This is especially true for coastal waters near industrially developed cities or cites of intensive agricultural activity, which are powerful local sources of nutrients into the atmosphere, and then to the surface of the sea with atmospheric precipitation. Therefore, the lack of data on the flows of nutrients with atmospheric precipitation may lead to an incomplete understanding of the anthropogenic impact on the nutrients cycle, incorrect assessment of the magnitude, variability, and biogeochemical consequences of the input of nitrogen, phosphorus, and silicon compounds of anthropogenic origin into the marine environment.

Since the Black Sea is one of the largest inland seas, it plays a significant role in the regional economy of coastal states for fishing, marine tourism, oil production, etc. Its coastal areas are also exceptionally recreational. At the same time, the coastal areas of the Black Sea are vulnerable to anthropogenic impact, including an increased input of nutrients. This results in an increase in the production of phytoplankton, fish, and shellfish [7,8]. The atmospheric input is expectedly a significant source of nutrients for the seashore of urbanized and industrial regions. In addition to background deposition of nutrients, local sources of nutrients to the atmosphere [9] can be also significant to dramatically increase atmospheric deposition of nutrients to the near-shore areas. This supports additional eutrophication (an increase in the input of organic matter to the system [10] and results in additional consumption of oxygen leading in hypoxia and even anoxia in coastal areas.

By bringing new nutrients to the upper layers of the ocean, aerosol deposition influences ocean productivity, especially in waters with low nutrient and chlorophyll concentration [11]. Krishnamurthy et al. (2010) [11] have demonstrated that atmospheric nitrogen input can have a significant impact on marine biogeochemistry at the regional scale and this impact can be increased by human activities. At the same time, atmospheric inputs of Si and P can have only minimal impact on the productivity and biogeochemistry of marine ecosystems, since these inputs are usually quite small compared to the flux of these nutrients from below the euphotic zone. Richon et al. (2017) [12] and Kanakidou et al. [13] have demonstrated that atmospheric deposition of nitrogen and dust phosphorus may potentially increase primary productivity over the Mediterranean Sea by 30–50% of the net primary production.

Results of our studies [14–16] have revealed both inter- and intra-annual variations in the load of nutrients with atmospheric depositions. We have revealed intra-annual variations in the inorganic nitrogen volume-weighted mean concentration with increased

values in winter and a decrease from March to August. The inorganic phosphorus and silicon volume-weighted mean concentrations are usually higher for the warm period of the year [16]. The inter-annual variations are less regular but concentrations and loads of nutrients generally increase on the scale of years. The ratio of nutrients in atmospheric depositions is even more complex because the major sources of nitrogen, phosphorus and silicon are different.

The ratio of nutrient concentrations in seawater may affect the elemental composition of particulate organic matter (POM) controlling phytoplankton population structure [17]. The main nutrients that cause eutrophication are nitrogen and phosphorus. Silicon is essential for the growth of diatoms. Anthropogenic activity has led to a sharp decrease in the silicon content in the Black Sea [18]. The effect of changes in the load and ration of nutrients with atmospheric depositions to the Black Sea remains unclear.

This work aims to study the input of inorganic nitrogen, phosphorus, and silicon with atmospheric depositions to coastal area near Crimea, variations in this input on the time scale of months-to-years, drivers of these variations, the range of changes in the ratio of nutrients, and effects of atmospheric input of nutrients on primary production and potential oxygen consumption on a daily-time scale.

## 2. Materials and Methods

### 2.1. Monitoring Sites

Two sampling stations at the coast of Crimea (Figure 1) have been positioned to monitor and collect atmospheric depositions in Katsiveli (a rural site at the Southern coast of Crimea) and in Sevastopol (a large industrial city at the Western coast of Crimea).

Katsiveli is a rural site located off large industrial cities and major highways. Katsiveli is located on Crimea's southern shore. Its population is about 500–600 people, but it increases seasonally by up to 20-fold because Katsively is primarily a vacationing summer place. There is no other business but tourism and leisure. Thus, this is typical site to monitor background characteristics of atmospheric depositions [https://en.wikipedia.org/wiki/Katsiveli, (accessed on 3 December 2021)]. Sevastopol is the largest city and the major seaport of Crimea. It is one of the most industrially developed cities in Crimea with intensive navy and maritime activities. Its population is under 500,000 but might increase twofold on summertime. Thus, anthropogenic and industrial pressures are very typical for this location [https://en.wikipedia.org/wiki/Sevastopol, (accessed on 3 December 2021)].

About 70% of samples have been collected in both locations from October to March and about 30%–from April to September.

### 2.2. Analytical Methods

Dry depositions of aerosol particles and contribution of their soluble components to wet depositions can significantly influence amount of nutrients in samples, thus in calculated estimates. To assess contribution of dry deposition of nitrogen, phosphorus and silicon, permanently open and wet-only samplers have been used in parallel since 2015 in Sevastopol and since 2016 in Katsiveli. Unlike the bulk samples, collected in open sampler, wet-only samplers are closed with a lid when there is no precipitation. Therefore, in principle, there is no effect of dry deposition on the collected sample [19]. A polyethylene bucket was used to collect rain samples.

Collected samples were photometrically analyzed in the laboratory in Marine Hydrophysical Institute RAS (Sevastopol) shortly after sampling or were frozen after sampling for further analyses. Ammonium was determined using a modified Sadgi–Solorzano method [20], which is based on determining the indophenolic dye forming in an alkaline medium from phenol, ammonia, and hypochlorite. The method uses nitroprusside as a catalyst for the reaction. The sensitivity of this method is 0.05 $\mu$M. To determine the nitrate-nitrite concentration, we used the method of reducing nitrates to nitrites with copper-bonded cadmium on a flowing system of AutoAnalyzer. Minimum detectable nitrates are 0.36 $\mu$M with the method error is $\pm$0.20 $\mu$M [21]. The concentration of silicon

was determined based on the formation of a blue silicomolybdenum complex, during procedure a correction for the salinity was applied. The determining the inorganic phosphorus concentration was carried out by the method based on the formation of a blue phosphoromolybdenum complex.

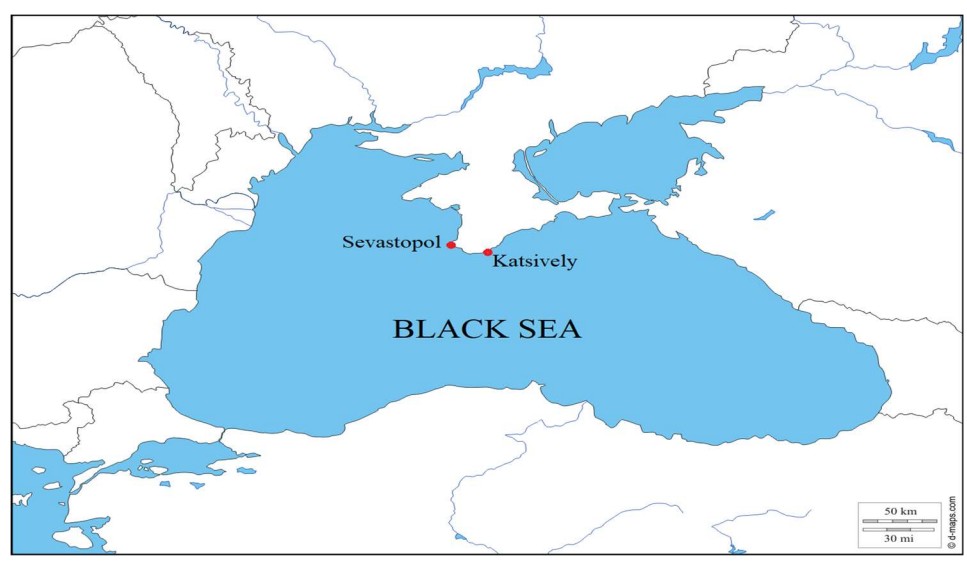

(a)

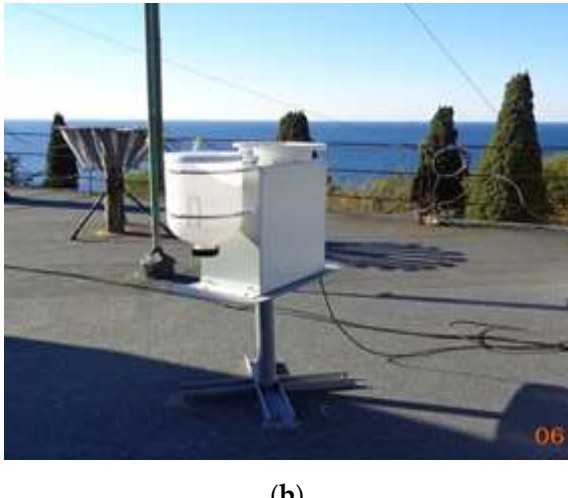

(b)

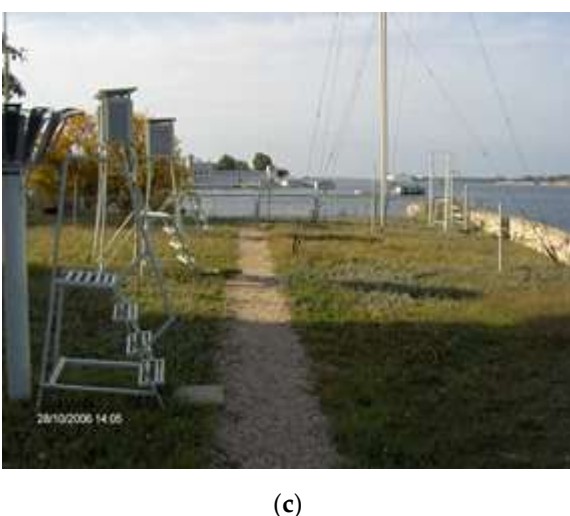

(c)

**Figure 1.** Monitoring sites: layout of sampling stations (**a**), station in Katsiveli (**b**), station inSevastopol (**c**).

Hydrometeorological data that are independent characteristics of the state of environment have been also recorded during the work. These data are used for interpretation the dynamics of the nutrients content in the atmosphere.

Statistical data processing has been performed by descriptive statistics methods, correlation (correlation coefficient and correlation ratio) and regression (paired and multiple regression) analyses using the Statistica program (StatSoft, Inc., Tulsa, OK, USA) and "Data analysis" add-on in Excel. The significance of correlation coefficients, correlation relations, and determination coefficients was checked by the Student's criterion. The Kolmogorov–Smirnov criterion was used to assess the normality of empirical distributions. The critical value of the significance level is assumed to be 0.05.

To exclude errors, the rule of 3 sigma ($3\sigma$) was applied, which allows to exclude data whose deviation from the arithmetic mean exceeds the tripled standard deviation. As a result, three values of the total concentration of inorganic forms of nitrogen obtained

during the analyses of precipitation samples in Sevastopol were discarded and five values for Katsiveli results.

## 3. Results

We have collected over 400 samples of atmospheric depositions in both sampling locations in 2015–2020. Some statistical characteristics of nutrient's concentrations are presented in the Tables 1 and 2.

**Table 1.** Characteristics of nutrients concentration in the sample of atmospheric depositions in Sevastopol.

|  | Nitrogen | | Phosphorus | | Silicon | |
|---|---|---|---|---|---|---|
|  | Wet-Only Sampler | Open Sampler | Wet-Only Sampler | Open Sampler | Wet-Only Sampler | Open Sampler |
| Amount of data | 469 | 437 | 459 | 429 | 461 | 432 |
| Max, μM | 465.76 | 799.58 | 18.17 | 37.12 | 34.46 | 36.79 |
| Min, μM | 19.50 | 17.00 | 0 | 0 | 0 | 0 |
| VWM, μM | 71.48 | 87.57 | 0.38 | 0.91 | 0.78 | 2 |
| St. deviation, μM | 65.99 | 98.00 | 1.7 | 3.5 | 3.1 | 5.1 |

**Table 2.** Characteristics of nutrients concentration in the sample of atmospheric depositions in Katsiveli.

|  | Nitrogen | | Phosphorus | | Silicon | |
|---|---|---|---|---|---|---|
|  | Wet-Only Sampler | Open Sampler | Wet-Only Sampler | Open Sampler | Wet-Only Sampler | Open Sampler |
| Amount of data | 254 | 499 | 385 | 354 | 387 | 357 |
| Max, μM | 391.80 | 461.80 | 19.94 | 9.15 | 4.96 | 15.53 |
| Min, μM | 17.00 | 21.45 | 0 | 0 | 0 | 0 |
| VWM, μM | 82.14 | 92.36 | 0.4 | 1.3 | 0.9 | 1.77 |
| St. deviation, μM | 55.29 | 69.01 | 0.77 | 2.15 | 0.70 | 2.09 |

The maximum inorganic nitrogen concentration in Sevastopol was detected in September 2019 and April 2019 for the wet-only and open samplers respectively. The maximum inorganic phosphorus concentration for wet-only sampler was detected in June 2017, and for the open one in July 2015. Maximum silicon concentration in the samples of atmospheric deposition collected in wet-only sampler was detected in September 2017. For the open sampler maximum silicon concentration was detected in November 2018. In Katsiveli the maximum inorganic nitrogen concentration was detected in December 2020 for both samplers. Maximum phosphorus concentration for wet-only sampler was detected in May 2015, and for the open one in July 2016. Maximum concentrations of silicon for both types of samplers were detected in June 2016.

Inorganic nitrogen in wet samples in both rural and urban locations was mainly represented by nitrates and ammonium (Table 3).

**Table 3.** Contributions of inorganic nitrogen forms for sampling sites.

| Sampling Site | Nitrates, % | Ammonium, % | Nitrites, % |
|---|---|---|---|
| Sevastopol | 57 | 42 | 1 |
| Katsiveli | 54 | 36 | 7 |

The intra-annual distribution of nutrients monthly volume-weighted mean concentrations (VWM) with the atmospheric precipitations is shown in Figures 2 and 3.

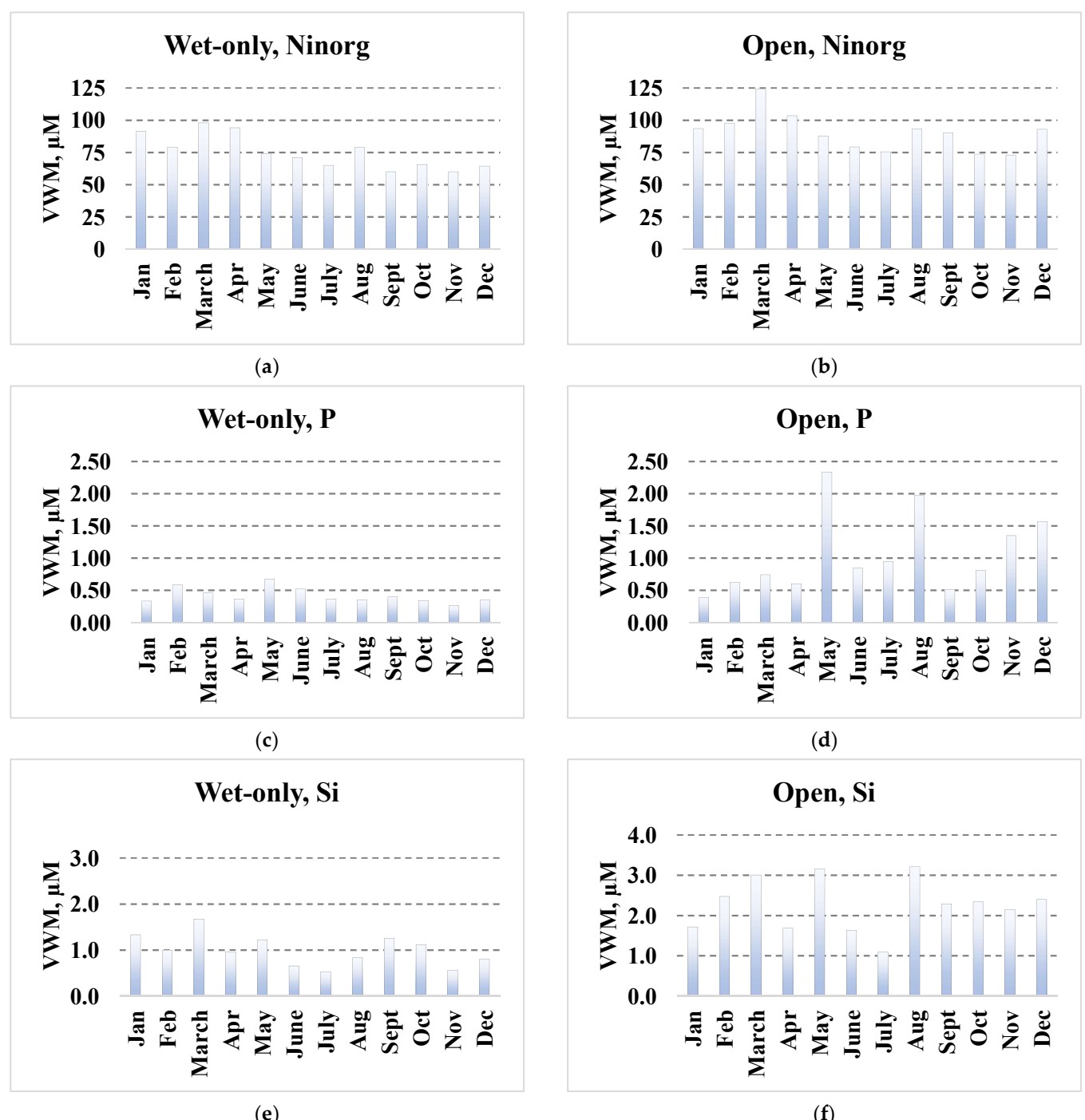

**Figure 2.** Intra-annual distribution of nutrients monthly volume-weighted mean concentrations in the atmospheric precipitations in Sevastopol in both types of samplers: for inorganic nitrogen (**a**,**b**), inorganic phosphorus (**c**,**d**), silicon (**e**,**f**).

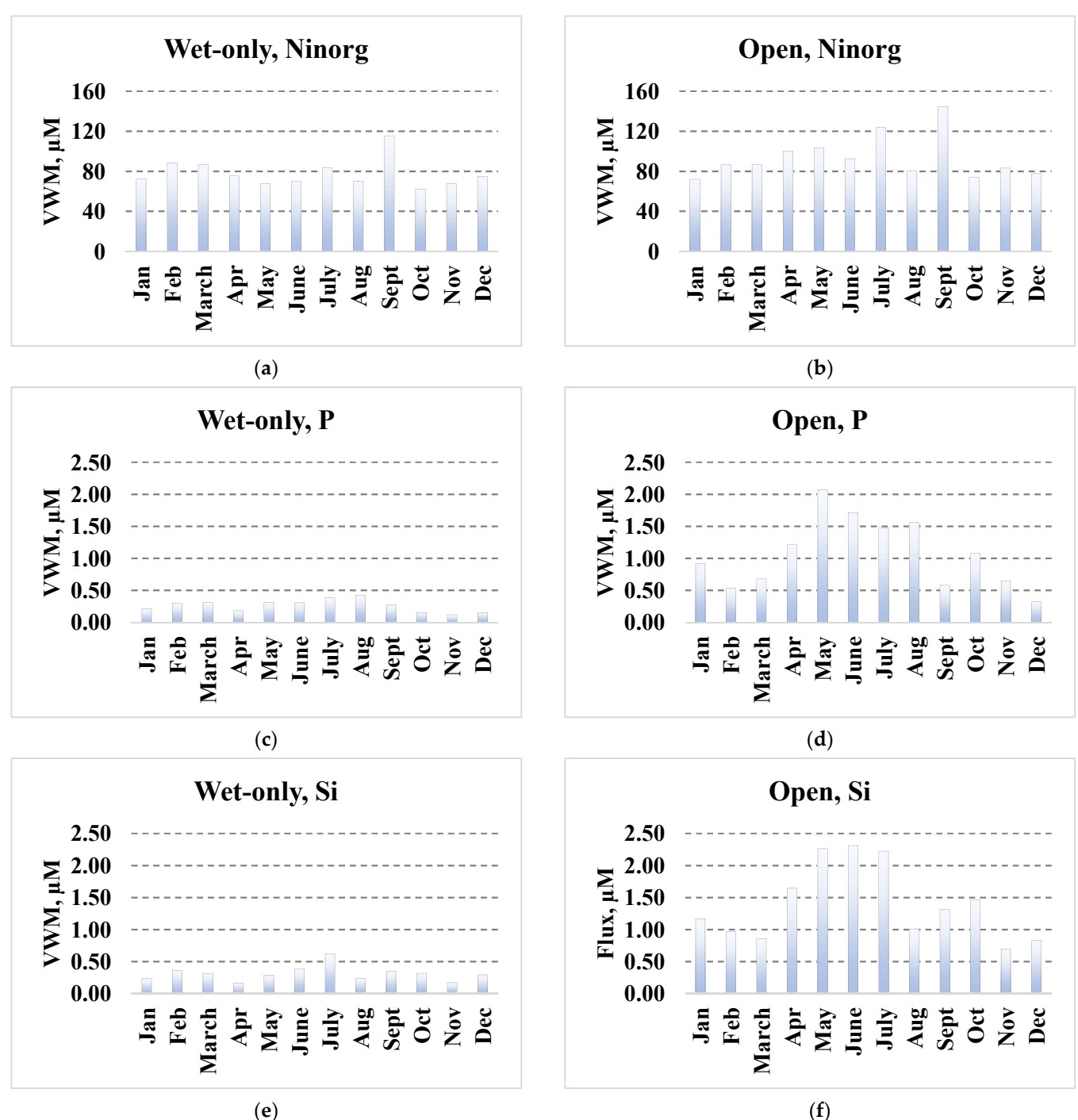

**Figure 3.** Intra-annual distribution of nutrients monthly volume-weighted mean concentrations in atmospheric precipitations in Katsiveli in both types of samplers: for inorganic nitrogen (**a**,**b**), inorganic phosphorus (**c**,**d**), silicon (**e**,**f**).

The molar phosphorus and silicon fluxes with atmospheric depositions in both urban and rural areas were significantly lower than inorganic nitrogen flux. Yet, there were recorded cases of significant increase in inorganic phosphorus and silicon concentrations reaching more than 300% in samples collected in samplers collected wet and dry depositions as compared to samples collected in samplers for wet depositions (Figure 4).

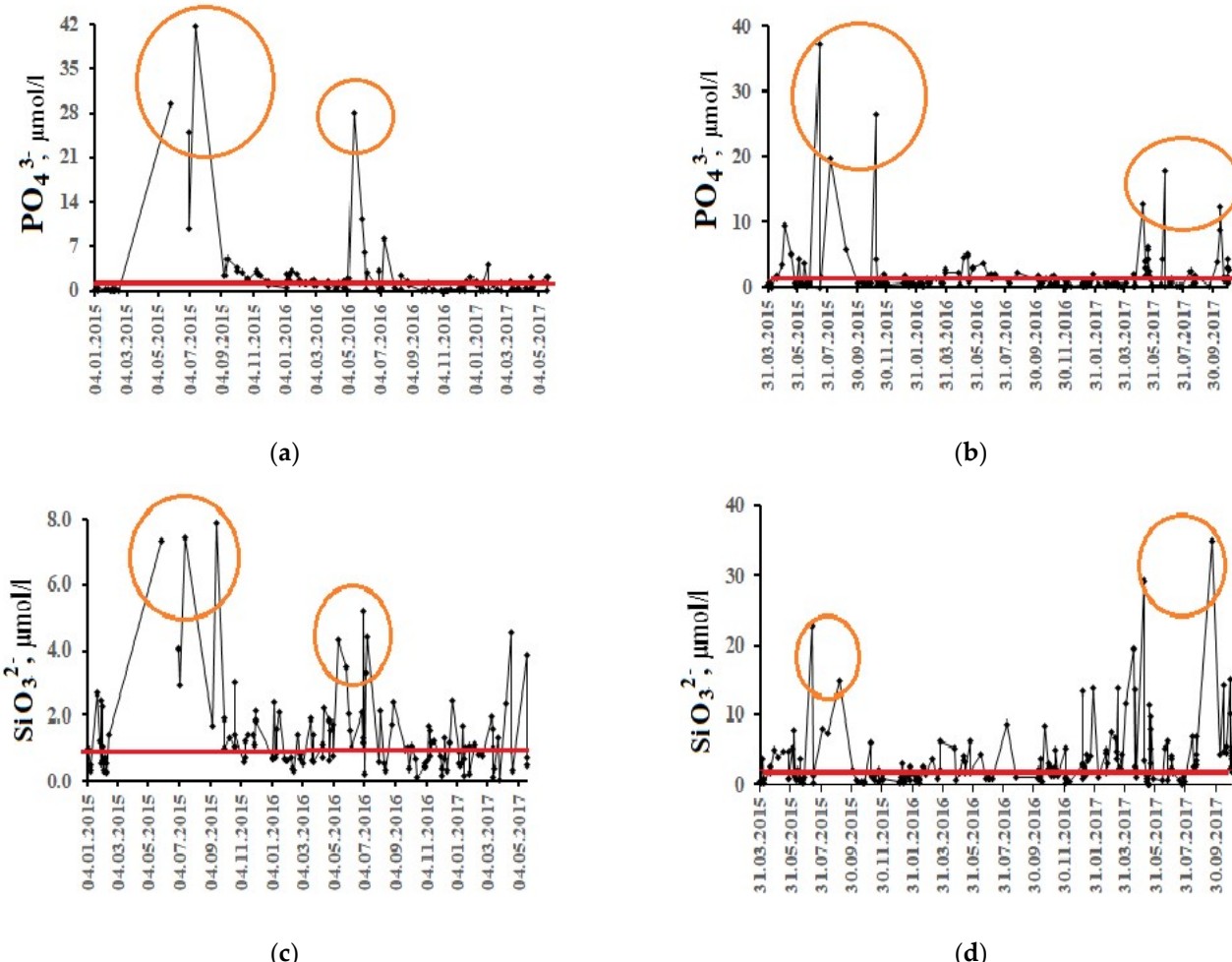

**Figure 4.** Variations of inorganic phosphorus (**a**,**b**) and silicon (**c**,**d**) concentrations in the atmospheric depositions in Katsiveli (**a**,**c**) and Sevastopol (**b**,**d**). Red line—volume-weighted mean concentration of nutrients, orange ovals—cases of significant excess of inorganic phosphorus and silicon concentrations.

Some of these cases are for the same dates in both locations supporting the fact that dust is brought by one air mass to Sevastopol and Katsiveli. We have previously demonstrated [22] that these cases (equal and/or different in dates for Sevastopol and Katsiveli) can trace the influence of dust transport with air masses from Sahara and Syria deserts and depends on regional precipitation and atmospheric conditions.

## 4. Discussion

The formation of the chemical composition of precipitation depends on various factors, first of all, the content of various substances in the atmospheric air. The input of nutrients into the atmosphere can be natural or/and anthropogenic. The presence of nitrogen in the air is often associated with anthropogenic sources. Industrial production of nitric acid, fuel burning, motor vehicle exhaust gas and further development of transport are the main sources of nitrogen oxide emissions. About 90% of ammonium emissions come from agriculture and industrial ammonia production [23]. The main natural sources of phosphorus in the atmosphere are wind erosion of soil in areas with arid climates; generation of biogenic aerosols by terrestrial vegetation (spores, pollen, plant residues, etc.); destruction of sea spray on the surface of reservoirs, as well as volcanic activity. Anthropogenic phosphorus input is associated with the production and utilization of fertilizers, as well as with metallurgical industries [5]. Silicon in the atmosphere is associated with airborne particles from industrial areas or transported with dust from deserts. Global dust emissions range

from 1000 to 3000 Mt/yr [24]. Then all these nutrients are transported from the atmosphere to the underlying surface with atmospheric depositions.

Many researchers [25–27] argued that dry depositions contribute equal or greater amounts of nutrients, as compared to wet depositions. When it comes to silicon, it is in the form of lithogenic silica and silicates and it is transported with airborne particles, originated from industrial burning and/or soil particles [28]. Thus, the content of silicon in samples from open samplers for wet and dry precipitations may dramatically exceed the content of silicon in samples from wet-only samplers. A similar situation is for phosphorus in the form of phosphate [25,27,29]. The average contribution of dry depositions is about 208% for phosphorus, while it is about 405% for silicon. These values agree with data [30], where the concentration of inorganic phosphorus in dry depositions are on average 2.8-times higher than that in wet precipitations. Yet, nitrogen is usually in the form of dissolved and gaseous compounds, thus contribution of dry deposition of nitrogen is less important, as compared to silicon and phosphorus. Indeed, the average contribution of dry depositions for inorganic nitrogen is about 22–26% at Sevastopol (Table 1) and 17–21% at Katsiveli (Table 2) in 2015–2020. Our results on nitrogen deposition are in line with the results [29], which have assessed contributions of wet and dry depositions to the total amount of NOx are similar.

In previous studies [15], we have demonstrated that the monthly volume weighted mean concentrations of inorganic nitrogen in atmospheric depositions in Sevastopol (the urban site) increase in the cold season (Figure 5a). We have assumed that this increase is a result of burning fuel (gas, coal, oil) to produce heat in October–March. Furthermore, there is a clear difference between intra-annual variations of nitrogen concentration in Sevastopol and Katsiveli. These oscillations of the monthly volume weighted mean concentrations in rural site are statistically insignificant (Figure 5b). We have explained this fact by the absence of powerful local sources of air pollution (neither agricultural, nor industrial) in Katsiveli. Thus, the situation with powerful local sourced and seasonal variations in inorganic nitrogen in Sevastopol has been recognized as clear and reasonable.

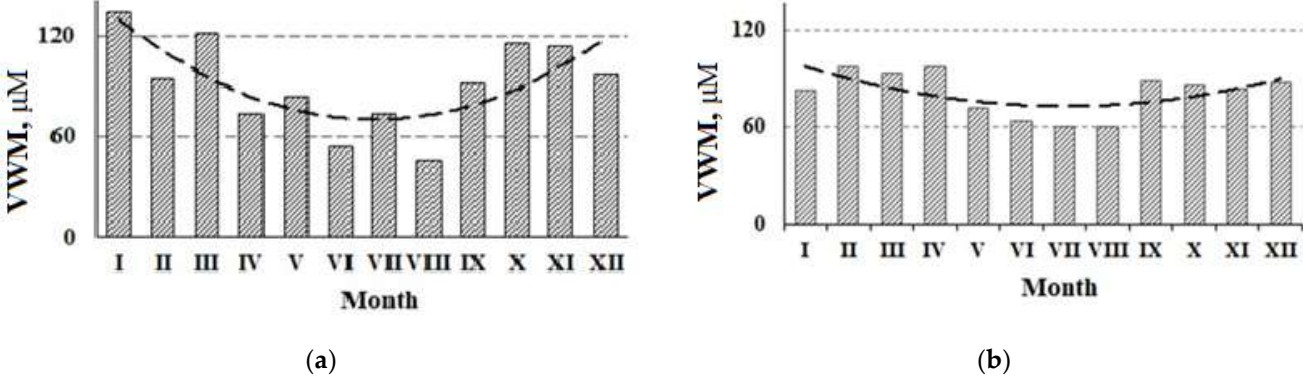

(a)　　　　　　　　　　　　　　　　　　　　(b)

**Figure 5.** Intra-annual variations of the inorganic nitrogen volume-weighted mean concentrations in the atmospheric depositions in Sevastopol (**a**) and Katsiveli (**b**) in 2007–2008.

Surprisingly, the results obtained in 2015–2020 demonstrate no statistically significant intra-annual variations in the concentration of inorganic nitrogen in atmospheric depositions in both locations in this period (Figures 2 and 3). To make the situation even more complicated, there was observed a slightly (but statistically insignificant) increase in concentrations of inorganic nitrogen in Katsiveli in the warm period of year (Figure 3b).

There are several possible reasons of these dramatic changes in seasonal variations of inorganic nitrogen volume-weighted mean concentrations. They can result from significant changes in the volume of anthropogenic emissions to the atmosphere as compared to the period before 2014, that is, an increase in the quantity of marine, aviation and other

transportation in Crimea, especially in summer. For example, the number of auto-vehicles in Sevastopol increased twofold from 2015 to 2017. The number of large city boiler houses in Sevastopol increased from 116 in 2014 to 133 in 2016, but these boilers utilized natural gas instead of coal. This suggests qualitative and quantitative changes in local anthropogenic sources of inorganic nitrogen to atmospheric depositions.

Intra-annual variations of volume-weighted mean concentrations (excluding cases of extremely high concentrations) of inorganic phosphorus reveal no statistically significant variations (Figure 2c,d) in Sevastopol, while these values increase in Katsiveli in April-August (Figure 3c,d).

Considering that phosphorus is mostly of lithogenic origin in the atmosphere, this difference between Sevastopol and Katsiveli is due to different sources of dust in the air. Anthropogenic activities, transportation, and excavations are intensive in Sevastopol in all seasons providing an even background level of the phosphorus input to the air and on average 0.42 μM of inorganic phosphorus in atmospheric depositions. Anthropogenic activities in Katsiveli are far less intensive, as compared to Sevastopol, and a background concentration of phosphorus in the atmospheric depositions for low-seasons is about 0.2 μM, but it increases in the high-season due to the several-fold multiplying population and anthropogenic activities in summer. Additionally, a dramatic decrease of precipitation events in summer period and windy weather conditions support an additional source of soil dust to the air and dry atmospheric depositions.

The concentration of silicon in atmospheric depositions dramatically increases in Katsiveli on summertime (Figure 3f). This behavior of silicon is similar to phosphorus. We assume that the reasons for this intra-annual variations in the concentration of silicon are also similar to those for phosphorus. Silicon is a lithogenic element and a dramatic decrease in rain precipitation in Crimea in summertime and additional input of silicon to the atmosphere in dry windy conditions result in the observed high concentrations of this element in atmospheric depositions even though extreme events of long-distance transport of dust from the Sahara and Syria deserts are excluded. If high summertime concentrations of silicon are excluded, the average concentration of silicon in the area of Katsiveli is as low as 0.3 μM, which is typical for natural coastal conditions.

The average concentration of silicon in atmospheric depositions in Sevastopol is several-fold of that value in Katsiveli (Figures 2 and 3). Similar to phosphorus, this demonstrates the presence of powerful local anthropogenic sources of this element to the air and then to the atmospheric depositions.

The Redfield ratio is often used to assess the effect of nutrients (inorganic nitrogen, phosphorus) input with atmospheric depositions on primary production [11,30–32]. The ratio of produced organic carbon and utilized nitrogen, phosphorus, and silicon in marine ecosystems (C:N:P:Si) is equal to 106:16:1:15 [33].

The N:P ratio is specifically variable for atmospheric depositions and may influence the Redfield's ratio of carbon (C), nitrogen (N), phosphorus (P), and oxygen ($O_2$) in marine ecosystems. The Redfield's C:N:P:$O_2$ ratio equals 106:16:1:138. Koçak et al. (2016) [34] have argued that atmospheric P and N depositions can support new production from 1.9 to 5.7 and from 1.5 to 5.4 mgC·m$^{-2}$·day$^{-1}$, respectively. Increased precipitation of ammonium and nitrate promotes increased dissolved inorganic nitrogen in surface waters and water acidification. If the ratio of nitrogen to phosphorus or silicon is sufficiently shifted towards nitrogen, limitation of primary production for diatoms is expected. These N:P:Si proportions determine limitation for phytoplankton growing and could thus affect the community composition and ecosystem processes [35].

Our data reveal several distinct features in seasonal and site-to-site variations in the N:P:Si ratios (Table 4).

Seasonal variations in this ratio are far more pronounced for rural Katsiveli, as compared to industrial Sevastopol. This suggests that natural variations are larger and depend on long-distance atmospheric transport of nitrogen, phosphorus, and silicon from different remote sources and local weather conditions supporting either wet or dry atmospheric

depositions. Unlike for rural Katsiveli, powerful local sources of nitrogen, phosphorus, and silicon equal or even override natural sources of these elements, resulting in rather stable intra-annual ratios.

**Table 4.** VWM seasonal and average for 2015–2020 values of the N:P:Si ratio in atmospheric depositions in Sevastopol and Katsiveli normalized for phosphorus values.

|           | Sevastopol | Katsiveli |
|-----------|------------|-----------|
| Winter    | 186:1:2.4  | 340:1:1.3 |
| Spring    | 161:1:2.4  | 239:1:0.9 |
| Summer    | 161:1:1.5  | 220:1:1.4 |
| Autumn    | 185:1:3.0  | 454:1:1.6 |
| 2015–2020 | 172:1:2.3  | 297:1:1.3 |

The ratio N:P in the atmospheric depositions is significantly shifted towards nitrogen in both locations, as compared to the Redfield ratio. This result is consistent with data published by Markaki et al. [36]. The ratio P:Si is significantly shifted towards phosphorus comparing to the Redfield ratio. Thus, atmospheric depositions support the traced shift in the phytoplankton succession from diatoms to coccolithophores and further to dinoflagellates supporting intensive "red tides" and oxygen deficit in the coastal area.

There is a difference in the contribution of nitrogen in these ratios in Sevastopol and Katsiveli. It is about twofold higher in Katsiveli, as compared Sevastopol (Table 4). However, this is due to lower phosphorus concentrations in the atmospheric depositions in Katsiveli and about twofold higher concentrations in Sevastopol. This is also true for silicon, which is depleted comparing to nitrogen and phosphorus, and far more depleted in Katsiveli comparing to Sevastopol.

The situation is dramatically different for rare atmospheric depositions with extremely high concentrations of phosphorus and silicon (Table 5). For these events, the N:P:Si ratio is phosphorus enriched and nitrogen depleted. Yet even these phosphorus and silicon enriched precipitations support silicon depleted ratios in seawaters near both Katsiveli and Sevastopol. Unlike the usual atmospheric depositions (Table 4), these phosphorus enriched depositions support shifts from dinoflagellates to coccolithophores and partially explain extreme bloom of Emiliania huxleyi in the Black Sea in June. Still, atmospheric depositions cannot support a desired shift from non-diatoms to diatoms in the Black Sea.

**Table 5.** N:P:Si ratio for the cases with extremely high concentrations P in the atmospheric depositions.

| Date         | Sevastopol | Katsiveli  |
|--------------|------------|------------|
| 24 June 2017 | 11:1:0.3   | 17:1:0.2   |
| 29 May 2015  | 6.6:1:0.4  | 11:1:0.2   |
| 15 July 2015 | 6.2:1:0.6  | 6.3:1:0.2  |
| 13 May 2016  | 23:1:0.8   | 13.5:1:0.2 |
| 28 May 2017  | 3:1:0.5    | 10.6:1:0.3 |

A comparative analysis of the nutrients content in the sea surface layer before and immediately after precipitation for the surface layer of the Sevastopol Bay near the site of sampling atmospheric depositions has revealed that the value of the nutrients concentrations in the surface layer was higher than the background concentration after precipitation [37]. Depending on the trophic status the water regions may react differently to the input of nutrients with precipitation. For example, in the most unproductive areas nutrients can be consumed very quickly and do not lead to an increase in their concentration in the surface layer. Our results show that after precipitation, the concentration of inorganic phosphorus in the surface layer of the Sevastopol Bay may increase over fivefold compared to the background concentration. The increase in the silicon concentration can reach 30%.

The input of precipitations containing a significant amount of nutrients into the bay can significantly change the Redfield ratio in surface layer (Table 6).

**Table 6.** N:P:Si ratio in the surface layer of the Sevastopol Bay.

| Season | N:P | Si:P | N:P | Si:P |
|---|---|---|---|---|
| | **Background** | | **After Deposition** | |
| Spring | 129 | 62 | 326 | 90 |
| Summer | 59 | 38 | 134 | 119 |
| Autumn | 78 | 48 | 248 | 61 |
| Winter | 188 | 51 | 217 | 64 |

This shift can reach three times the background value, especially during the warm stratification period. A change in the ratio towards one or another nutrient, in turn, can cause the flowering of some phytoplankton species and limit the development of others.

Applying the Redfield C:N ratio we can estimate the effect of precipitation. Berthold et al. [38] have studied the amount and influence of atmospheric phosphorus deposition to the Baltic Sea. The median daily deposition of phosphate has been estimated by 56 $\mu$g m$^{-2}$ day$^{-1}$ (1.8 $\mu$mol m$^{-2}$ day$^{-1}$), the median annual phosphate deposition has leveled 16.7 kg km$^{-2}$ a$^{-1}$. The median TP-depositional rates have ranged between 19 and 70 kg km$^{-2}$ a$^{-1}$. The highest TP-depositional rates have been measured on summer. Gao et al. [39] have estimated the total atmospheric inorganic nitrogen deposition to the South China Sea. The authors conclude that this flux is comparable to large riverine inputs and potentially contributes 1.8–11.1% of nitrogen to new production and 0.7–1.8% of nitrogen to primary production.

The average annual input of inorganic nitrogen with atmospheric precipitations in Sevastopol region according to data obtained in this work is 31.1 mmol·m$^{-2}$·year$^{-1}$. This may support consumption of 205.9 mmol·m$^{-2}$·year$^{-1}$ of inorganic carbon. According to [40], the average annual primary production in coastal areas is 100–130 gC·m$^{-2}$·year$^{-1}$. Based on these values, the average annual nitrogen input with atmospheric precipitation can lead to an increase in primary production by 1.9–2.5%. The annual input of dissolved inorganic phosphorus with atmospheric precipitation (0.33 mmol·m$^{-2}$·year$^{-1}$) corresponds to an additional carbon production of 35 mmol·m$^{-2}$·year$^{-1}$. This contributes to an increase in primary production in the coastal areas of the Black Sea by ~0.4% of its long-term average value. The additional supply of silicon with atmospheric precipitation (0.78 mmol·m$^{-2}$·year$^{-1}$) contributes to additional organic carbon production of 5.51 mmol·m$^{-2}$·year$^{-1}$, which is a very small amount. The maximum effect on the value of primary production is due to the input of inorganic nitrogen with precipitation.

For this possible increase in primary production (205.9 mmol·m$^{-2}$·year$^{-1}$), we have estimated the amount of oxygen that will be required for its oxidation. Using the C:O ratio equal to 106:138, we have found that additional oxygen consumption during mineralization of organic matter can reach 268 mmol·m$^{-2}$·year$^{-1}$. The annual input of inorganic nitrogen with atmospheric precipitation can lead to insignificant (about 1.5%) additional oxygen consumption in the surface water layer. However, precipitation is not a permanent phenomenon, does not fall evenly throughout the year. That is why we assumed their effect on a shorter (daily) timescale. The maximum nitrogen input of 3.6 mmol·m$^{-2}$·day$^{-1}$ has been determined in December. According to [41], the winter period accounts for about 40% of the annual primary production in coastal areas. In conversion on the daily values of primary production, a single event of rain can lead to additional oxygen consumption for the oxidation of organic matter in an amount ~78%.

## 5. Conclusions

Atmospheric depositions of inorganic nitrogen in urban and rural locations at the Crimean coast of the Black Sea have been mainly presented by nitrate and ammonium. Nitrates have prevailed in the urban site of Sevastopol, while ammonium prevails in the rural cite of Katsiveli.

There have been no statistically significant intra-annual variations in the nitrogen volume-weighted mean concentrations in Sevastopol in 2015–2020 (unlike period 2007–2008). On the contrary, increasing concentrations of nitrogen have been detected in Katsiveli in warm period. Probably it can be explained by some changes in local nutrients sources since 2014 in the Crimea.

The phosphorus and silicon fluxes with atmospheric depositions in both urban and rural stations are significantly lower than the inorganic nitrogen flux. However, there have been recorded cases of a significant excess of inorganic phosphorus and silicon concentrations in samples collected in wet and dry samplers reaching more than 300% of those collected in wet-only samplers.

In both locations, the N:P:Si ratio in the atmospheric depositions has been significantly shifted towards nitrogen as compared to the stoichiometric ratio. However, in cases of extremely high concentrations of phosphorus and silicon, the role of these nutrients may become more important for the surface layer of the sea.

Our results suggest that additional flux of nutrients with atmospheric depositions may change the C:N:P:Si ratio in the surface layer of the Black Sea. Moreover, the input of the atmospheric depositions with extremely high concentrations of silicon and phosphorus (even episodic events) can affect the marine ecosystem, including biological community and chlorophyll-a concentration. The effect of nutrients input with the atmospheric depositions can lead to additional oxygen consumption for the oxidation of organic matter in an amount practically corresponding to its daily production during photosynthesis supporting oxygen deficit and suboxic conditions.

Thus, the results of this work demonstrate that atmospheric depositions of inorganic nitrogen, phosphorus and silicon are of great importance for even the coastal marine system of the Black Sea.

**Author Contributions:** A.V.V.—formulation of scientific hypotheses; chemical analysis of the atmospheric precipitations samples, writing—original draft preparation; interpretation of the results obtained. S.K.K.—scientific supervision of research. All authors have read and agreed to the published version of the manuscript.

**Funding:** This work was carried out within the framework of the state assignments of Marine Hydrophysical Institute of RAS No. 0555-2021-0004 and Russian Foundation for Basic Research (RFBR) according to the research project No. 19-05-00140.

**Institutional Review Board Statement:** Not applicable.

**Informed Consent Statement:** Not applicable.

**Data Availability Statement:** All the data and additional information supporting the findings of this study are available from the corresponding authors upon reasonable request.

**Conflicts of Interest:** The authors declare no conflict of interest.

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
