# Peer review of "Variations in Concentrations and Ratio of Soluble Forms of Nutrients in Atmospheric Depositions and Effects for Marine Coastal Areas of Crimea, Black Sea"

_applsci, doi:10.3390/app112311509_

Round 1
Reviewer 1 Report
Dear authors,
Congratulation for submitting the manuscript. Your article seems well written and with a lot of data. I also appreciated that you've conducted the study on many years which demonstrate attention and patience.
Regarding the structure of the article, I have some recommendations as follows:
- The aim of the study is not clearly presented. You said that you wanted to study the input of P and N but didn't mention the final purposes of this study. What do you want to achieve by those studies.
 - The results are not statistically interpreted even it is mentioned in the Material and method section
 - Material and method section does not present enough information: the tests that were used for statistical interpretation (if none were applied, you can not say that the results are significant or not); the devices that were used to extract the data, etc.
 - Some of the results are presented in the Results section (these results are not interpreted) while others are included in the discussion section (why?).
 - The introduction is too long, the interpretation of results is missing and the Discussion section is to short.

regarding the content, I have some comments and questions:
- Where din you presented the following data: "the average contribution of dry depositions 246 for inorganic nitrogen is about 22–26% at Sevastopol and 17-21% at Katsiveli in 2015-2020."
 - when is the worm season? because on the paragraph from line 271- fig 3b shows the highest value in September (not in July), and for 2b, the highest values appear in March and April.
 - lines 299-303 - how do you know that the value of Si in atmosphere is influenced by the dust brought from Sahara or Syria?
 - lines 327-331 - how you compared the values from both locations, you could apply statistical interpretation to see is the differences between them are significant or not.
 - table 4 - you are discussing about C and C:N, but the table presents ratios of N:P and Si: P.

I hope my comments will help you to improve your article.

Reviewer 2 Report
Varenik and Konovalov measured and reported the variations of nutrients’ concentrations and their ratios (N:P:Si) in aerosols collected from Black Sea close to Crimea area. They found that dry deposition contributes more Si and P than wet deposition. Also, their results show significant difference in the N:P:Si ratio between the aerosols and the Redfield ratio. They proposed that the input of nutrients by aerosols may alter the local ratio of nutrients and thus affect the alteration of ecosystems.
The observations seem to be convincing and systematic. The emphasis on the delivery of N, P, and Si by the aerosols to the near-shore aquatic systems is novel. However, the major proposition (the nutrient ratio may be largely affected by the aerosol input) is not convincing depending on the current results. One major issue is that the authors ignored riverine transport as another potentially significant or more important source of nutrients in the seashore systems. Also, whenever compared with ecological ratios (the Redfield ratio, for example), they should always use the biologically available component (for P, Si, and N).
Major comment:
You should use the biologically available phases of N, P, and Si when compared the N:P:Si ratio in aerosols to the Redfield ratio used in the ecological studies. This is because a large proportion of P and Si are not biologically available, i.e., they don’t contribute to the ecosystems.
Also, to evaluate the influence of aerosol input on the N:P:Si ratio in the surface waters, you need to make sure the relative contribution of riverine transport vs. aerosol input of “biologically available” nutrients. I see that the sampling sites are close to the land, so the riverine system may play the predominant roles in supplying nutrients.
Minor:
There are multiple cases of grammar mistakes. I suggest the author to carefully check the text.
Line 115: “is comparable to”
Line 134: “P:SiO2”
Line 143: “This work aims to”
Line 168: “can significantly influence amount of”
Line 250: “the concentration of inorganic phosphorus in dry depositions is on average…”
Line 252: “who” should be “which”
Line 278: “has increased” should be “increased”; “utilize” should be “utilized”
Line 301: “results in” should be “result in”.
Line 302: “even” should be “even though”
Line 325: “equal” should be “is equal to”
Line 330 – 332: here and elsewhere (e.g., lines 343 – 347 and 359 – 362), how about the riverine transport of nutrients?
Round 2
Reviewer 1 Report
Thank you for your answers and for considering my suggestions.
Reviewer 2 Report
Thanks for considering my comments!